# Axl, Immune Checkpoint Molecules and HIF Inhibitors from the Culture Broth of *Lepista luscina*

**DOI:** 10.3390/molecules27248925

**Published:** 2022-12-15

**Authors:** Mihaya Kotajima, Jae-Hoon Choi, Mitsuru Kondo, Corina N. D’Alessandro-Gabazza, Masaaki Toda, Taro Yasuma, Esteban C. Gabazza, Yukihiro Miwa, Chiho Shoda, Deokho Lee, Ayaka Nakai, Toshihide Kurihara, Jing Wu, Hirofumi Hirai, Hirokazu Kawagishi

**Affiliations:** 1Graduate School of Science and Technology, Shizuoka University, 836 Ohya, Suruga-ku, Shizuoka 422-8529, Japan; 2Research Institute of Green Science and Technology, Shizuoka University, 836 Ohya, Suruga-ku, Shizuoka 422-8529, Japan; 3Faculty of Agriculture, Shizuoka University, 836 Ohya, Suruga-ku, Shizuoka 422-8529, Japan; 4Research Institute for Mushroom Science, Shizuoka University, 836 Ohya, Suruga-ku, Shizuoka 422-8529, Japan; 5Department of Immunology, Graduate School of Medicine, Mie University, Edobashi 2-174, Tsu 524-8507, Japan; 6Department of Ophthalmology, Keio University School of Medicine, 35 Shina-nomachi, Shinjuku-ku, Tokyo 160-8582, Japan

**Keywords:** *Lepista luscina*, Axl, immune checkpoint molecules, hypoxia-inducible factor, tryptanthrin

## Abstract

Two compounds **1** and **2** were isolated from the culture broth of *Lepista luscina*. This is the first time that compound **1** was isolated from a natural source. The structure of compound **1** was identified via 1D and 2D NMR and HRESIMS data. Compounds **1** and **2** along with 8-nitrotryptanthrin (**4**) were evaluated for their biological activities using the A549 lung cancer cell line. As a result, **1** and **2** inhibited the expression of Axl and immune checkpoint molecules. In addition, compounds **1**, **2** and **4** were tested for HIF inhibitory activity. Compound **2** demonstrated statistically significant HIF inhibitory effects on NIH3T3 cells and **1** and **2** against ARPE19 cells.

## 1. Introduction

Natural products are an important source for valuable compounds with unique action mechanisms because of their immense structural diversity and wide variety of biological activities. There are approximately 1.5 million species of fungi as complex and diverse as plants and animals on the earth, of which only about 10% have been named and described. Fungi that form fruiting bodies have been attracting attention, because they produce secondary metabolites with various pharmaceutical and biological activities. 

Lung cancer is the leading cause of cancer morbidity and mortality worldwide. Therefore, molecular target drug discovery and drug development for lung cancer are actively carried out worldwide. Axl, a receptor-type tyrosine kinase, has been designated as a leading candidate for targeted cancer therapy [1]. Programmed cell death ligands 1 (PD-L1) and 2 (PD-L2) have been clinically reported as promising targets in cancer treatment [1,2]. Similarly, blocking of immune checkpoint molecules has caused clinical responses in lung cancer patients in a recent clinical study [1]. There is ongoing research to explore the possibility of cancer therapeutics that target these signaling pathways (Axl and immune checkpoint molecules). 

On the other hand, vascular diseases of the retina, including diabetic retinopathy and age-related macular degeneration, are the leading cause of vision loss worldwide [3]. It is well known that overexpression of vascular endothelial growth factor (VEGF) plays a central role in the pathogenesis of these diseases. To date, anti-VEGF therapy has been widely used to treat pathologic neovascularization of the retina [4]. Hypoxia-inducible factor (HIF) serves as a strong transcription regulator of VEGF induction under hypoxic and other stress conditions [5]. HIF inhibitors, from anti-cancer drugs to natural products, suppressed retinal neovascularization and ectopic VEGF expression in murine models [6,7,8,9]. Therefore, isolation of cancer and vascular disease therapeutics from natural products may also contribute to drug discovery.

The genus of one fruiting-body-forming fungus, *Lepista* sp. (Tricholomataceae), is widespread, with many edible species such as *L. sordida* and *L. nuda.* Approximately 60 species have been described in this genus, which is widely distributed in Asia, North America and Europe [10]. *Lepista* sp. is also known to form fairy rings, which are common in grasslands. “Fairy rings” are zones in which grass growth is stimulated and/or suppressed by the interaction of fungi and plants worldwide [11]. In the previous studies, we found a plant growth promoter, 2-azahypoxanthine (AHX) and a plant growth inhibitor, imidazole-4-carboxamide (ICA), from the culture broth of *L. sordida* mycelia as causative substances of fairy rings by the fungus [12,13]. Subsequently, 2-aza-8-oxohypoxanthine (AOH) was discovered from AHX-treated rice as a metabolite of AHX [14], and the three compounds were named “fairy chemicals” (FCs) after the article in Nature in 2014 that covered our chemical research [15]. Lee et al. demonstrated that AHX has an inhibitory effect on HIF activation in retinal cells [7]. AOH is effective as a cosmetic skin barrier against water loss [16,17]. ICA suppressed the expression of Axl and immune checkpoint molecules (PD-L1/PD-L2) [18]. *L. nuda* is reported to exhibit many biological activities such as antioxidant, antimicrobial and cytotoxic activity [19,20]. The extracts of *L. inversa* demonstrated cytotoxicity against various cancer cell lines [21] and clitocine was identified as the active compound from *L. inversa* [22]. On the other hand, no research has been performed on bioactive substances from our target in this study, *L. luscina*. 

As mentioned above, we have recently been focusing on the search for Axl and PD-L1/PD-L2 inhibitors and the inhibitors of HIF activation from natural products, especially mushroom-forming fungi. Therefore, we attempted to isolate biologically active metabolites from liquid cultures of the unexplored fungus *L. luscina* and evaluate their bioactivities. In this study, we report the isolation of two compounds (Figure 1) and their inhibitory activities against Axl, immune checkpoint molecules and HIF luciferase.

## 2. Results and Discussion

### 2.1. Identification of Compounds

Compound **1** was isolated as an yellow powder with the molecular formula C_16_H_10_N_2_O_3_ based on the molecular ion peak at *m/z* 279.0787 [M + H]^+^ (calcd. for C_16_H_11_N_2_O_3_, 279.0764) in HRESIMS. The ^1^H- and ^13^C-NMR data { H-1 δ_H_ 8.43 (1H, dd, 8.0, 1.7), C-1 δ_C_ 127.4; H-2 δ_H_ 7.67 (1H, ddd, 8.0, 8.2, 1.1), C-2 δ_C_ 130.2; H-3 δ_H_ 7.84 (1H, ddd, 8.2, 8.1, 1.7), C-3 δ_C_ 134.9; H-4 δ_H_ 8.02 (1H, br d, 8.0), C-4 δ_C_ 130.7} in combination with the COSY correlations (H-1/H-2, H-2/H-3, H-3/H-4) and HMBC correlations (H-1/C-3, C-4a, C-12; H-2/C-3, C-4, C-12a; H-3/C-1, C-4, C-4a; H-4/C-2) together with an amido carbon (C-12) at δ_C_ 157.8 and an imino carbon (C-5a) at δ_C_ 144.8 revealed the presence of a quinazolinone moiety (Figure 2 and Appendix A and Table 1). The ^1^H- and ^13^C-NMR data {H-9 δ_H_ 7.31 (1H, d, 9.2, 2.5), C-9 δ_C_ 125.1; H-7 δ_H_ 7.38 (1H, d, 2.2), C-7 δ_C_ 108.4; H-10 δ_H_ 8.52 (1H, d, 9.2), C-10 δ_C_ 119.2} along with a carbonyl group (C-6) at δ_C_ 182.7 indicated the existence of a 1,2,4-trisubstituted benzoyl moiety. The moiety confirmed COSY (H-9/H-10) and HMBC correlations (H-7/C-6, C-8, C-9, C-10a; H-9/C-7, C-10a; H-10/C-6a, C-8, C-10a). The presence of a methoxy group at C-8 was suggested by ^1^H-NMR (δ_H_ 3.90, s, 3H) and HMBC correlation (-OCH_3_/C-8). Although no HMBC correlations were observed between the two moieties, the position of the linkage of them was determined via the chemical shift of C-10a (δ_C_ 140.5). As a result, this compound was identified as 8-hydroxyindolo [2,1-*b*]quinazoline-6,12-dione (**1**). This compound has already been synthesized; however, this was the first reported isolation of **1** from a natural source [23].

Compound **2** was identified to be indolo [2,1-*b*]quinazoline-6,12-dione, tryptanthrin, via interpretation of MS, NMR data and X-ray crystallography analysis (Table 2 and Appendix A). Compound **2** is a well-known alkaloid with an indolequinazolinone structure, and mainly exists in blue plants such as *Strobilanthes cusia* and indigo; however, this is the first isolation of **2** from a mushroom-forming fungus [24,25,26,27]. In addition, the alkaloid **2** has been isolated from the yeast *Candida lipolytica,* which proliferated when grown in L-tryptophan-containing medium [28]. Some studies revealed that **2** and its derivatives have a wide range of biological activities, such as anti-inflammatory activity [29], insecticidal activity [30], anti-allergy activity [31], a protective effect on colitis [32] and anti-bacterial activities [33]. Very recently, antiviral properties of **2** against HCoV-NL63 infection were reported [34]. Compound **2** is also an active ingredient of traditional Japanese herbal remedies for fungal infections [24]. Recently, there has been a lot of research on the anti-cancer effects of **2** [35,36,37,38,39,40,41]. The compound induces apoptosis of tumor cells [39,40] and inhibits the drug-resistance-related gene in breast and colon cancer cells [36,37]. It also reduces the activity of cyclooxygenase-2 [41].

### 2.2. Metabolic Pathway of Tryptanthrin (**2**) and Its Analogues (**1** and **3**)

High-performance liquid chromatography (HPLC) analysis with photodiode array (PDA) suggested the presence of the demethyl analogue of **1**, 8-methoxyindolo [2,1-*b*]quinazoline-6,12-dione (**3**), in the hexane soluble part of culture broth of *L. luscina*, although the compound could not be isolated due to the small amount of it (Appendix A). Recently, tandem mass spectrometry has been widely used for identification of various compounds due to its inherent accuracy and excellent sensitivity. Therefore, we attempted to identify the structure using LC-MS/MS as previously reported [42]. The MS spectrum of protonated tryptanthrin (**2**) (*m/z* 249) is shown in Figure 3. The MS/MS spectrum of protonated **2** shows fragment ions at *m/z* 130. Fragmentation on the quinazoline moiety produces ions at *m/z* 130 from the cleavage of the indole moiety [42]. The MS/MS spectra of protonated **1** and **3** show product ions at *m/z* 160 and 146, respectively. The ions at *m/z* 160 and 146 were found to be characteristic ions for the presence of a methoxy and a hydroxyl group at the phenyl groups of the indole moieties, respectively. All the data allowed us to conclude that compound **3** is 8-hydroxytryptanthirin.

Although there are numerous studies for the biological activities of tryptanthrin (**2**) and its derivatives, any biological activities of **1** remain unknown to date. Considering the oxidation process of aromatic rings, compound **1** should be biosynthesized from **2** via 8-hydroxytryptanthirin (**3**). Compound **3** was found as a metabolite of tryptanthrin (**2**) in rat and it was proposed that **3** might be formed from **2** by cytochrome P450 in rat liver microsomes [42]. However, this was the first reported detection of this compound from a source other than animals. In order to confirm the biosynthetic pathway of **1** from **2**, a CYP-inhibitor, 1-aminobenzotriazole, was added to the culture broth of *L. luscina*. As a result, the production of compounds **1** and **3** was inhibited and compound **2** was accumulated in the presence of the inhibitor (Figure 4). The result suggests that the CYP enzyme(s) is involved in the biosynthetic pathway.

### 2.3. Biological Activities

The compound 8-Nitroindolo [2,1-*b*]quinazoline-6,12-dione, 8-nitrotryptanthrin (**4**), has potent human indoleamine 2, 3-dioxygenase 2 (hIDO2) inhibitory activity [43], antitubercular activity [44] and antitrypanosomal activity [45]. In this study, we assessed the effects of compounds **1** and **2** along with their analogue (**4**) on the human A549 cell line. The cells were treated with each compound (**1**, **2** and **4**). Compounds **1** and **4** demonstrated cytotoxicity on the cells (data not shown). As shown in Figure 5, compound **2** inhibited expressions of all the genes (Axl, PD-L1 and PD-L2). Compounds **1** and **4** significantly suppressed the expression of Axl. Axl plays an important role in the epithelial–mesenchymal transition, which is an important step for the initiation of metastasis and development of resistance to drugs and chemotherapy [37]. The expression mechanisms between Axl and immune checkpoint molecules (PD-L1 and PD-L2) would be different from each other. However, a recent study found the correlation between Axl and immune checkpoint molecules in lung adenocarcinomas; Axl positively contributes to the expression of immune checkpoint molecules in the regulation of immune microenvironment and tumor proliferation [37]. Therefore, if the expression of these molecules was downregulated by a compound, the compound might become a promising candidate as an anticancer agent. The results in this study indicate that the tryptanthrin moiety without functional groups at C-8 played an important role in the suppression of PD-L1 and PD-L2 and tryptanthrin (**2**) was the most promising candidate for cancer therapy (Figure 5). There are only a few studies showing the presence of Axl and immune checkpoint molecules inhibitors in natural products. Recently, our group reported the isolation of compounds that decrease the expression of Axl and immune checkpoint molecules from the mushrooms *Leucopaxillus giganteus*, *Pleurocybella porrigens* and *Agaricus blazei* [46,47,48]. Our findings indicate that mushrooms are a potential source of natural Axl and PD-L1/PD-L2 inhibitors.

Compounds **1**, **2** and **4** were evaluated for HIF inhibitory activity via HIF luciferase assay (Figure 6). CoCl_2_ was used to induce HIF activation, and halofuginone and doxorubicin were used as positive controls for HIF inhibition. The NIH3T3 cell line was used because it is a widely used cell line for general understanding of the role of HIF and it is easy to develop a HIF-luciferase reporter stable cell line [49]. Compound **2** demonstrated significant inhibitory activity on HIF activation induced by CoCl_2_ in the cells. We also tested in vitro cell models for ophthalmic drug development such as the ARPE-19 cell line (human retinal pigmented epithelium cells) and 661W cell line (mouse immortalized cone photoreceptor cells with some features of retinal ganglion precursor-like cells) [50,51,52]. In ARPE-19 cells, compounds **1** and **2** demonstrated statistically significant HIF inhibitory effects. These results indicate that compound **2** (tryptanthrin) has the strongest inhibitory effect on HIF-activation among the three compounds.

In previous reports, **1** demonstrated higher anti-tobacco mosaic virus activity than that of **2** [23]. Furthermore, compounds **2** and **4** exhibited EC_50_ of 23.0 and 0.82 μM towards anti-trypanosomal activity, respectively [45]. Thus, tryptanthrin derivatives (**1** and **4**) displayed stronger biological activities than that of **2**. These derivatives possess a functional group (methoxy or nitro) at C-8 in the tryptanthrin moiety that seems to be important for their activities [23,45]. In contrast, in this study, tryptanthrin (**2**) demonstrated stronger activities in both the bioassays than **1** and **4,** which have an additional methoxy group and nitro group compared to **2**, respectively. These results indicate that both the electron-donating group and the electron-withdrawing one in **1** and **4** attached to **2** negatively affect the immune checkpoint- and HIF-inhibitory activities.

## 3. Materials and Methods

### 3.1. General Experimental Procedures

One- and two-dimensional ^1^H NMR spectra were recorded on a JNM-ECZ500R spectrometer at 500 MHz, and ^13^C NMR spectra were recorded on the same instrument at 125 MHz (JEOL, Tokyo, Japan). HRESIMS spectra were measured on a JMS-T100LP mass spectrometer (JEOL, Tokyo, Japan). HPLC separations were performed with a Jasco Chromatography Data Station ChromNAV system using reverse-phase HPLC columns (ODS-P, InertSustain, Tokyo, Japan). Silica gel plates (Merck F254), ODS gel plates (Merck F254) and silica gel 60 N (Kanto Chemical, Tokyo, Japan) were used for analytical TLC and flash column chromatography. All solvents used throughout the experiments were obtained from Kanto Chemical Co. (Tokyo, Japan).

### 3.2. Fungal Material

The mycelia of *L. luscina* (NBRC 31053) were pre-cultured on potato dextrose agar (PDA) to obtain the actively growing cultures, and the inoculated mycelia were incubated at 25 °C for a month. After growth, 10 pieces (6 mm diameter) cut from the four-week-cultured mycelia were inoculated into 500 mL Erlenmeyer flasks containing 300 mL of YG (0.3% yeast extract and 1% D-glucose) medium, and the cultures were incubated for 3 weeks (25 °C, 120 rpm). The cultivation was performed twice.

### 3.3. Extraction and Isolation

First culture broth (1.8 L) was filtered and evaporated under reduced pressure. The concentrate was divided into *n*-hexane, ethyl acetate (EtOAc) and water-soluble parts. The *n*-hexane-soluble part (5.0 mg) was separated via reverse-phase HPLC (OSP-P, 50% MeCN) to give compounds **1** (1.1 mg) and **2** (2.6 mg). Second culture broth (2.4 L) was also extracted using the same liquid–liquid extraction method, and the hexane soluble part (19.5 mg) was subjected to reverse-phase HPLC (OSP-P, 50% MeCN) to afford compounds **1** (2.2 mg) and **2** (5.0 mg).

### 3.4. X-ray Crystallography Analysis

Compound 2 was crystallized in hexane/CHCl_3_/MeOH. A single crystal was mounted on a MiTeGen loop with Paraton-N (Hampton Research, Aliso Viejo, CA, USA), and was flash frozen to 173 K in a liquid nitrogen cooled stream of nitrogen. Data collection was carried out on a Rigaku VariMax diffractometer using a multi-layer mirror monochromated Mo microfocus sealed X-ray source (1.2 kW) and a CCD detector. Data collection and reduction were performed using the CrysAlis PRO software package. The structure was solved via direct method and ShelXT, and refined using the SHELX97 tool [53,54]. All of the non-hydrogen atoms were refined anisotropically. The hydrogen atoms were placed in calculated positions and allowed to ride on the carrier atoms. Crystallographic data were deposited at the Cambridge Crystallographic Data Centre and allocated the deposition number CCDC 2216457. The data can be obtained free of charge via www.ccdc.cam.ac.uk/produts/csd/request (accessed on 31 October 2022). The size of the crystal used for measurement was 0.34 mm × 0.15 mm × 0.10 mm. Crystal data and the detailed structure determination of 2 are summarized in Appendix A. The structure analyzed was essentially the same with the paper previously reported [55].

### 3.5. Feeding Studies Using CYP-Inhibitor 1-Aminobenzotriazole

Two fungal pieces (6.0 mm diameter) were inoculated into each of two 100 mL Erlenmeyer flasks containing 30 mL of YG medium. Thirty microliters of 3 mM 1-aminobenzotriazole filtered through a membrane filter was added into the cultures, and the mycelia were incubated for 5 weeks (25 °C, 120 rpm). The amount of compound **1** in the culture broth was analyzed via LC-MS/MS.

### 3.6. LC-MS Spectrometry

LC-MS/MS analyses were performed with an UPLC system (Nihon Waters, Tokyo, Japan) coupled with a tandem Xevo TQ-S micro mass spectrometer (Nihon Waters, Tokyo, Japan). The UPLC mobile phases consisted of 0.1% formic acid in DW (A) and 0.1% formic acid in acetonitrile (B) (60% solvent B; flow rate, 0.2 mL/min; column oven, 40 °C, injection volume, 2 μL). The column used for the separation was an Inertsil ODS-4, 3 μm (*ϕ* 2.1 × 100 mm, GL Science). For **1** detection, MS analysis was performed in the positive mode with the following source parameters: cone voltage, 76 V; collision energy, 28 V (*m/z* 279 > 236). For **2** detection, MS analysis was performed in the positive mode with the following source parameters: cone voltage, 32 V; collision energy, 28 V (*m/z* 249 > 130). For **3** detection, MS analysis was performed in the positive mode with the following source parameters: cone voltage, 76 V; collision energy, 28 V (*m/z* 265 > 146). The following source parameters were common to all the compounds: capillary voltage, 3.0 kV; desolvation temperature, 500 °C; desolvation gas flow, 1000 L/h; cone gas flow, 50 L/h.

### 3.7. Axl and Immune Checkpoint Molecules Assay

The human A549 alveolar epithelial cell line was purchased from the American Type Culture Collection (Rockville, MD, USA) and cultured in DMEM, supplemented with 10% heat-inactivated fetal bovine serum, 2 mM L-glutamine and 100 U mL penicillin plus 100 U mL streptomycin. All cells were cultured at 37 °C in 75 cm^2^ flasks in an atmosphere composed of 5% CO_2_ and 95% air. Confluent cells were passaged after 5–7 days.

A549 cells in 0.1% BSA-DMEM were seeded in 24-well plates. Test compound (20 μg/mL) was added to the wells, and the plates were incubated for 24 h. Total RNA was extracted using Sepasol^®^-RNA I Super G (Nacalai, Kyoto, Japan) following the instructions of the manufacturer. One μg of total RNA was denatured at 65 °C for 10 min, and then reverse-transcribed using ReverTra Ace Reverse Transcriptase (TOYOBO, Osaka, Japan) and oligo (dT) primer in a volume of 20 μL according to the manufacturer’s protocol.

Each gene contains forward and reverse sequences (5′ > 3′) as GGAGCGAGATCCCTCCAAAAT and GGCTGTTGTCATACTTCTCATGG for the GADPH gene, TGCCATTGAGAGTCTAGCTGAC and TTAGCTCCCAGCACCGCGAC for the Axl gene, GGACAAGCAGTGACCATCAAG and CCCAGAATTACCAAGTGAGTCCT for the PD-L1 gene, and ACCGTGAAAGAGCCACTTTG and GCGACCCCATAGATGATTATGC for the PD-L2 gene, respectively. The cDNA was amplified using PCR and the conditions were as follows: 94 °C, 1 min; 60 °C, 1 min; and 72 °C, 1 min for 28–35 cycles. PCR products were electrophoresed on a 1.5% agarose gel and then stained with ethidium bromide solution. Semi-quantitative RT-PCR results were quantified by using ImageJ software.

Data are expressed as the mean ± standard error of the mean (SEM). The statistical difference was calculated via analysis of variance with *post hoc* analysis using Fisher’s predicted least significant difference test. All statistics were performed using the StatView 5.0 package (Abacus Concepts, Berkeley, CA, USA).

### 3.8. HIF Luciferase Assay

Murine cell lines of fibroblast NIH3T3 and cone photoreceptor 661W were cultured in DMEM (Cat #08456-36, Nacalai Tesque, Kyoto, Japan) media supplemented with 10% FBS and 1% streptomycin-penicillin at 37 °C under an atmosphere containing 5% CO_2_. A human cell line of retinal epithelial ARPE-19 was cultured in DMEM/F-12 (Cat #C11330500BT, Gibco, NY, USA) media with the same supplements as above. These cell lines were continuously maintained for in vitro experiments. A luciferase assay was performed as previously described [56]. Briefly, NIH3T3, 661W and ARPE-19 cell lines were transfected with a HIF-luciferase reporter gene construct (Cignal Lenti HIF Reporter, Qiagen, Venlo, The Netherlands). The HIF-luciferase construct encodes a firefly luciferase gene under the control of the HRE, which binds HIF. The cell lines were also co-transfected with a cytomegalovirus-renilla luciferase construct as an internal control and seeded at 1.0 × 10^4^ cells/well/70 µL (NIH3T3 and ARPE-19) or 0.8 × 10^4^ cells/well/70 µL (661W) in a white sterile HTS Transwell-96 receiver plate (Corning, Corning, NY, USA). At 24 h of cell stabilization, the cells were treated with CoCl_2_ (200 µM, cobalt (II) chloride hexahydrate, Wako, Saitama, Japan) to activate HIF. To evaluate the inhibitory effects of test compounds (1 mg/mL) against HIF activation, the cells were co-treated with each compound and CoCl_2_. After incubation for 24 h at 37 °C in a 5% CO_2_ incubator, luminescence was measured using a Dual-Luciferase Reporter Assay System (Promega, Madison, WI, USA). For the expected HIF inhibitory positive controls, 0.2 μM of halofuginone (Cayman Chemical, Ann Arbor, MI, USA) and 1 μM of doxorubicin (Tokyo Chemical Industry Co., Ltd., Tokyo, Japan) were used.

## 4. Conclusions

We isolated compounds **1** and **2** from the culture broth of *Lepista luscina* and detected **3** in the broth via LC-MS, and clarified that **1** is biosynthesized from **2** via 8-hydroxytryptanthirin (**3**) and the CYP enzyme(s) is involved in the biosynthetic pathway. Compound **2** inhibited the expression of Axl and immune checkpoint molecules. In addition, **2** demonstrated HIF inhibitory activity against NIH3T3 cells. Based on these findings, tryptanthrin (**2**), a potent inhibitor, can be used as a lead compound for the further development of therapeutic agents.

## Figures and Tables

**Figure 1 molecules-27-08925-f001:**
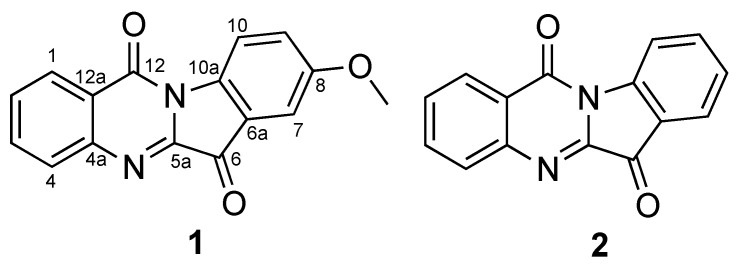
Structures of compounds **1** and **2**.

**Figure 2 molecules-27-08925-f002:**
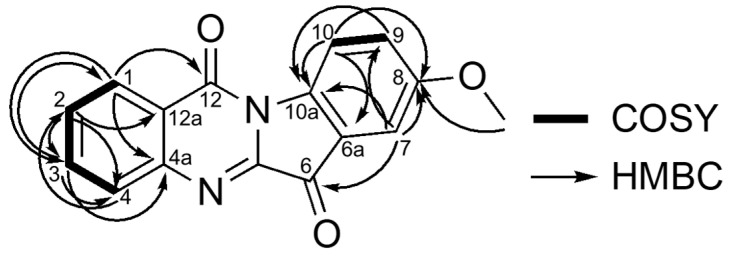
COSY and HMBC correlations of **1**.

**Figure 3 molecules-27-08925-f003:**
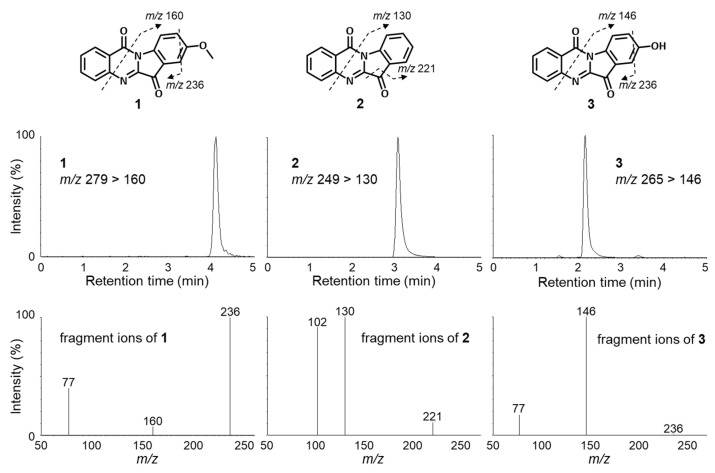
LC-MS/MS spectra (upper) and fragmentation (below) of compounds **1** to **3**.

**Figure 4 molecules-27-08925-f004:**
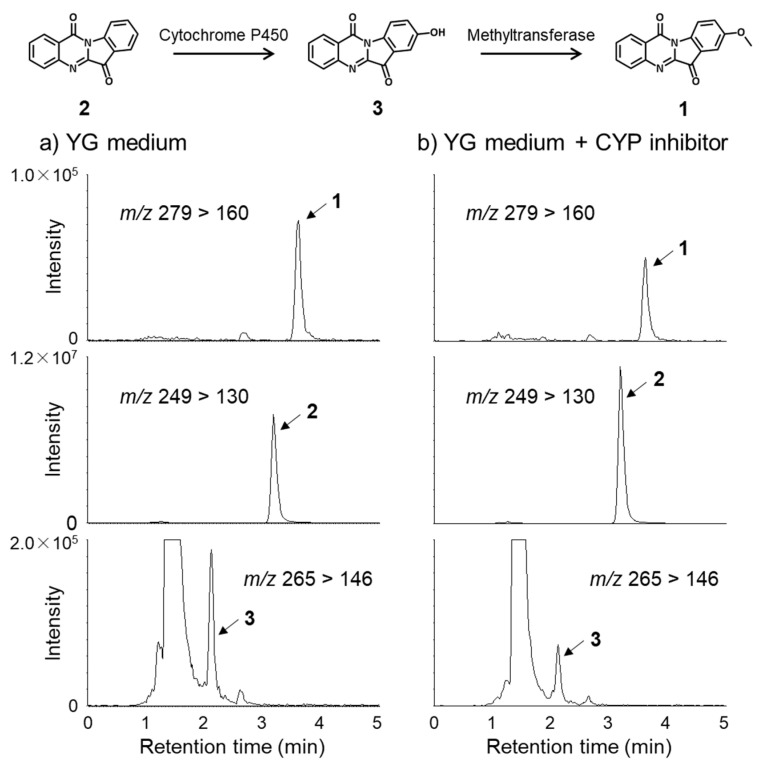
Comparison of production of compounds **1** to **3** without CYP inhibitor (**a**) to that with CYP inhibitor (**b**) in 4-week-cultured broth of *L. luscina*) via LC-MS.

**Figure 5 molecules-27-08925-f005:**
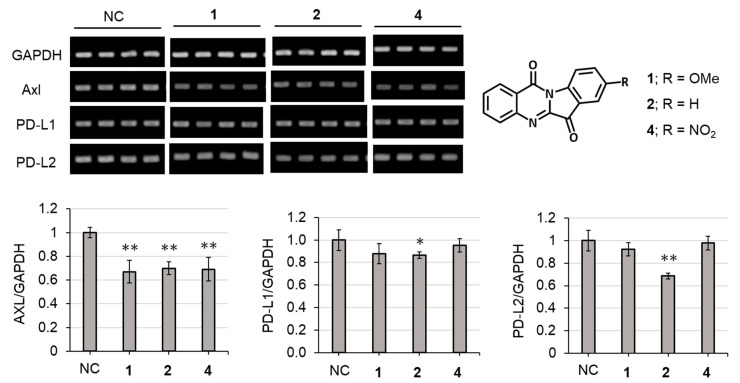
Effect of compounds **1**, **2** and **4** on expressions of Axl and immune checkpoint molecules (PD-L1 and PD-L2) in A549 cells. Values indicate means with standard deviation. NC indicates a negative control. Statistical analysis was performed using Fisher’s test (*: *p* < 0.05, **: *p* < 0.01 vs. NC, *n* = 4).

**Figure 6 molecules-27-08925-f006:**
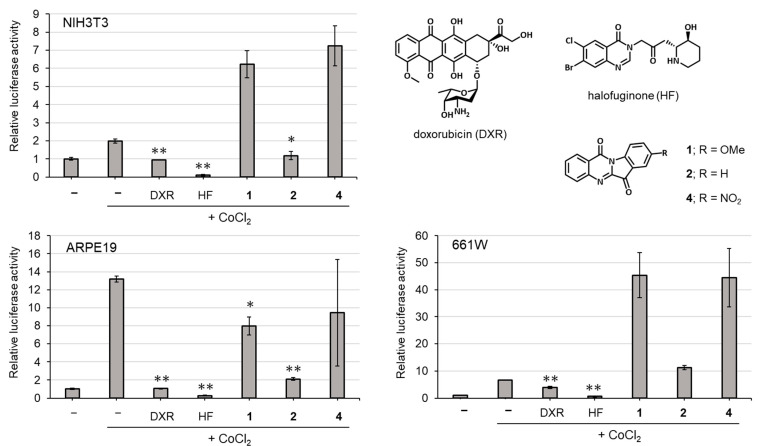
Inhibitory effects of compounds **1**, **2** and **4** on HIF activation. Values indicate means with standard deviation. Statistical analysis was performed using T-test (*: *p* < 0.01, **: *p* < 0.001 vs. CoCl_2_ treatment, *n* = 3). Doxorubicin (DXR) and halofuginone (HF) were used for HIF inhibitory positive controls.

**Table 1 molecules-27-08925-t001:** Data for **1** with regard to ^1^H and ^13^C NMR in CDCl_3_.

Position	δ_H_ (*J* in Hz)	δc
1	8.43 (dd, 8.0, 1.7)	127.4
2	7.67 (ddd, 8.0, 8.2, 1.1)	130.2
3	7.84 (ddd, 8.2, 8.1, 1.7)	134.9
4	8.02 (br d, 8.1)	130.7
4a	-	146.6
5a	-	144.8
6	-	182.7
6a	-	123.0
7	7.38 (d, 2.2)	108.4
8	-	158.8
9	7.31 (dd, 9.2, 2.9)	125.1
10	8.52 (d, 9.2)	119.2
10a	-	140.5
12	-	157.8
12a	-	123.9
-OCH_3_	3.90 (s)	56.0

**Table 2 molecules-27-08925-t002:** Comparison of ^13^C NMR data for compound **2** with those reported in CDCl_3_.

Position	Compound 2	Thyptanthrin [33]
	δc	δc
1	127.5	127.6
2	130.2	130.3
3	135.1	135.2
4	130.7	130.8
4a	146.6	146.7
5a	144.3	144.4
6	182.5	182.5
6a	121.9	122.0
7	125.4	125.4
8	127.2	127.2
9	138.3	138.3
10	118.0	118.0
10a	146.3	146.4
12	158.1	158.1
12a	123.7	123.8

## Data Availability

Not applicable.

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
