# Peer review of "Axl, Immune Checkpoint Molecules and HIF Inhibitors from the Culture Broth of Lepista luscina"

_molecules, 2022, doi:10.3390/molecules27248925_

Round 1
Reviewer 1 Report
The manuscript by Mihaya Kotajima et al. entitled ‘Axl, immune checkpoint molecules and HIF inhibitors from 2 the culture broth of Lepista luscina’ presents the isolation of tryptanthrin and its methoxy substituted analog from the culture broth of Lepista luscina. These compounds were detected in the broth by LC-MS. It was found that methoxy analog is biosynthesized from tryptanthrin in the presence of the CYP enzymes. Both isolated compounds inhibited the expression of Axl and immune checkpoint molecules. In addition, tryptanthrin showed HIF inhibitory activity against NIH3T3 cells, and ARPE19 cells. The described compounds constitute promising candidates as antitumor drugs. The experiments have been carried out carefully and correctly analyzed.
The reviewer has the following comments:
1. It is recommended that the description of compounds 1, 2, 3, and 4 should be given their chemical names as follows: indolo[2,1-b]quinazoline-6,12-dione; 8-methoxyindolo[2,1-b]quinazoline-6,12-dione, 8-hydroxyindolo[2,1-b]quinazoline-6,12-dione, and 8-nitroindolo[2,1-b]quinazoline-6,12-dione
2. It should be considered changing the numeration of compounds. Having in mind that compound 1 constitutes an isolated methoxy-derivative of tryptanthrin, the numbering should be reversed: i.e. for tryptanthrin - indolo[2,1-b]quinazoline-6,12-dione (1) and its derivatives:
8-hydroxyindolo[2,1-b]quinazoline-6,12-dione (2), and 8-methoxyindolo[2,1-b]quinazoline-6,12-dione (3)
3. A detailed spectroscopic description for compounds (lines 93 to 108 and line 125 - including Tables 1 and 2) should be moved to the experimental part.
4. The introduction section (Paragraph 1) should be revised. It would be worth adding a deeper discussion on the mechanisms of biological activity of tryptanthrin.
5. Line 114: „Compound 2 is a well-known…” and the following phrases concerning on biological activity of tryptanthrin should be moved from Paragraph 2. (Results and Discussion) to Paragraph 1. (Introduction).
6. Line 177: Change the „anticancer reagent” into „anticancer agent”.
Reviewer 2 Report
The work presented by the authors shows the identification of three compounds, two of them isolated from Lepista luscina cultures.
The authors in the introduction describe the antitumor activity of some compounds with a similar structure. However, there is no clear relationship between anticancer activity and retinal diseases.
the results and discussions are well described. However, there is a typographical error in Figure 5; the third figure should say PD-L2.
On the other hand, the structural relationship with the biological activity of compounds must be described in greater detail and describe or propose structural modifications to obtain compounds with greater activity.
In relation to the conclusions, these do not really exist; rather it is a summary of the results already exposed. It is suggested to conclude regarding the structure of isolated and elucidated compounds for the generation of compounds with biological activity.
